# Minimum Divergence Estimators, Maximum Likelihood and the Generalized Bootstrap

**DOI:** 10.3390/e23020185

**Published:** 2021-01-31

**Authors:** Michel Broniatowski

**Affiliations:** Faculté de Mathématiques, Laboratoire de Probabilité, Statistique et Modélisation, Université Pierre et Marie Curie (Sorbonne Université), 4 Place Jussieu, CEDEX 05, 75252 Paris, France; michel.broniatowski@sorbonne-universite.fr

**Keywords:** statistical divergences, minimum divergence estimator, maximum likelihood, bootstrap, conditional limit theorem, Bahadur efficiency

## Abstract

This paper states that most commonly used minimum divergence estimators are MLEs for suited generalized bootstrapped sampling schemes. Optimality in the sense of Bahadur for associated tests of fit under such sampling is considered.

## 1. Motivation and Context

Divergences between probability measures are widely used in statistics and data science in order to perform inference under models of various kinds; parametric or semi-parametric, or even in non-parametric settings. The corresponding methods extend the likelihood paradigm and insert inference in some minimum “distance” framing, which provides a convenient description for the properties of the resulting estimators and tests, under the model or under misspecification. Furthermore, they pave the way to a large number of competitive methods, which allows to trade-off between efficiency and robustness, among other things. Many families of such divergences have been proposed, some of them stemming from classical statistics (such as the Chi-square divergence), while others have their origin in other fields, such as information theory. Some measures of discrepancy involve regularity of the corresponding probability measures while others seem to be restricted to measures on finite or countable spaces, at least when using them as inferential tools, henceforth in situations when the elements of a model have to be confronted with a dataset. The choice of a specific discrepancy measure in specific context is somehow arbitrary in many cases, although the resulting conclusion of the inference might differ accordingly, above all under misspecification.

The goal of this paper is explained shortly. The current literature on risks, seen from a statistical standpoint, has developed in two main directions, from basic definitions and principles, following the seminal papers [1,2].

A first stream of papers aims to describe classes of discrepancy indices (divergences) associated with invariance under classes of transformations and similar properties; see [3,4,5] for a review.

The second flow aims at making use of these indices for practical purposes under various models, from parametric models to semi-parametric ones, mostly. Also the literature in learning procedures makes extensive use of divergence-based risks, with a strong accent on the implementation issues. Following the standard approach, their properties are mainly considered under i.i.d. sampling, providing limit results, confidence areas, etc; see [6,7] and references therein for review and developments, and the monographs [8,9]. Also comparison among discrepancy indices are considered in terms of performances either under the model, or with respect to robustness (aiming at minimizing the role of outliers in the inference by providing estimators with redescending influence function), or with respect to misspecification, hence focusing on the loss in estimation or testing with respect to the distance from the assumed model to the true one.

This literature, however, rarely considers the rationale for specific choices of indices in relation with the concepts which define statistics, such as the Bayesian paradigm or the maximum likelihood (ML) one; for a contribution in this direction for inference in models defined by linear constraints, see [10]. In [11], we could prove that minimum divergence estimators (in the class of the ones considered in the present paper) coincide with MLEs under i.i.d. sampling in regular exponential models (but need not, even in common models such as mixtures). Here it is proved that minimum divergence estimators are indeed MLEs under weighted sampling, instead of standard i.i.d. one, commonly met in bootstrap procedures which aim at providing finite sample properties of estimators through simulation.

This paper considers a specific class of divergences, which contains most of the classical inferential tools, and which is indexed by a single scalar parameter. This class of divergences belongs to the Csiszar-Ali-Silvey-Arimoto family of divergences (see [4]), and is usually referred to as the power divergence class, which has been considered by Cressie and Read [12]; however this denomination is also shared by other discrepancy measures of some different nature [13]. We will use the acronym CR for the class of divergences under consideration in this paper.

Section 2 recalls that the MLE is obtained as a proxy of the minimizer of the Kullback-Leibler divergence between the generic law of the observed variable and the model, which is the large deviation limit for the empirical distribution. This limit statement is nothing but the continuation of the classical ML paradigm, namely to make the dataset more “probable” under the fitted distribution in the model, or, equivalently, to fit the most “likely” distribution in the model to the dataset.

Section 3 states that given a divergence pseudo distance ϕ in CR the Minimum Divergence Estimator (MDE) is obtained as a proxy of the minimizer of the large deviation limit for some bootstrap version of the empirical distribution, which establishes that the MDE is MLE for bootstrapped samples defined in relation with the divergence. This fact is based on the strong relation which associates to any CR ϕ-divergence a specific RV *W* (see Section 1.1.2); this link is the cornerstone for the interpretation of the minimum ϕ-divergence estimators as MLEs for specific bootstrapped sampling schemes where *W* has a prominent rôle. Some specific remark explores the link between MDE and MLE in exponential families. As a by product, we also introduce a bootstrapped estimator of the divergence pseudo-distance ϕ between the distribution of the data and the model.

In Section 4, we specify the bootstrapped estimator of the divergence which can be used in order to perform an optimal test of fit. Due to the type of asymptotics handled in this paper, optimality is studied in terms of Bahadur efficiency. It is shown that tests of fit based on such estimators enjoy Bahadur optimality with respect to other bootstrap plans when the bootstrap is performed under the distribution associated with the divergence criterion itself.

The discussion held in this paper pertains to parametric estimation in a model $P_{\Theta }$ whose elements $P_{\theta }$ are probability measures defined on the same finite space $Y:=\left\{d_{1},\ldots ,d_{K}\right\}$, and θ∈Θ is an index space; we assume identifiability, namely different values of θ induce different probability laws $P_{\theta }$’s. Also all the entries of $P_{\theta }$ will be positive for all θ in Θ.

### 1.1. Notation

#### 1.1.1. Divergences

We consider regular *divergence functions*
φ which are non negative convex functions with values in $\bar{R^{+}}$ which belong to $C^{2}$$\left(R\right)$ and satisfy $\varphi \left(1\right)=\varphi^{\prime }\left(1\right)=0$ and $\varphi^{\prime \prime }\left(1\right)=1$; see [3,4] for properties and extensions. An important class of such functions is defined through the power divergence functions
(1)$$\varphi_{\gamma }\left(x\right):=\frac{x^{\gamma }-\gamma x+\gamma -1}{\gamma \left(\gamma -1\right)}$$
defined for all real γ≠0,1 with $\varphi_{0}\left(x\right):=-\log x+x-1$ (the likelihood divergence function) and $\varphi_{1}\left(x\right):=x\log x-x+1$ (the Kullback-Leibler divergence function). This class is usually referred to as the Cressie-Read family of divergence functions (see [12]). It is a very simple class of functions (with the limits in γ→0,1) which allows to represent nearly all commonly used statistical criterions. Parametric inference in commonly met situations including continuous models or some non-regular models can be performed with them; see [6]. The $L_{1}$ divergence function $\varphi \left(x\right):=\left|x-1\right|$ is not captured by the CR family of functions. When undefined the function φ is declared to assume value +∞.

Associated with a divergence function φ,
ϕ is the *divergence* between a probability measure and a finite signed measure; see [14].

For $P:=\left(p_{1},\ldots ,p_{K}\right)$ and $Q:=\left(q_{1},\ldots ,q_{K}\right)$ in $S^{K},$ the simplex of all probability measures on Y, define, whenever *Q* and *P* have non-null entries
$$\phi \left(Q,P\right):=\sum_{k=1}^{K}p_{k}\varphi \left(\frac{q_{k}}{p_{k}}\right).$$

Indexing this pseudo-distance by γ and using $\varphi_{\gamma }$ as divergence function yields the Kullback-Leibler divergence KL(Q,P):=$\phi_{1}\left(Q,P\right):=\sum q_{k}\log\left(\frac{q_{k}}{p_{k}}\right)$, the likelihood or modified Kullback-Leibler divergence
$$KL_{m}\left(Q,P\right):=\phi_{0}\left(Q,P\right):=-\sum p_{k}\log\left(\frac{q_{k}}{p_{k}}\right),$$
the Hellinger divergence
$$\phi_{1/2}\left(Q,P\right):=\frac{1}{2}\sum p_{k}{\left(\sqrt{\frac{q_{k}}{p_{k}}}-1\right)}^{2},$$
the modified (or Neyman) $\chi^{2}$ divergence
$$\chi_{m}^{2}\left(Q,P\right):=\phi_{-1}\left(Q,P\right):=\frac{1}{2}\sum p_{k}{\left(\frac{q_{k}}{p_{k}}-1\right)}^{2}{\left(\frac{q_{k}}{p_{k}}\right)}^{-1}.$$

The $\;\chi^{2}$ divergence
$$\phi_{2}\left(Q,P\right):=\frac{1}{2}\sum p_{k}{\left(\frac{q_{k}}{p_{k}}-1\right)}^{2}$$
is defined between signed measures; see [15] for definitions in more general setting, and [6] for the advantage to extend the definition to possibly signed measures in the context of parametric inference for non-regular models. Also the present discussion which is restricted to finite spaces Y can be extended to general spaces.

The conjugate divergence function of φ is defined through
(2)$$\tilde{\varphi }\left(x\right):=x\varphi \left(\frac{1}{x}\right)$$
and the corresponding divergence $\tilde{\phi }\left(P,Q\right)$ is
$$\tilde{\phi }\left(P,Q\right):=\sum_{k=1}^{K}q_{k}\tilde{\varphi }\left(\frac{p_{k}}{q_{k}}\right)$$
which satisfies
$$\tilde{\phi }\left(P,Q\right)=\phi \left(Q,P\right)$$
whenever defined, and equals +∞ otherwise. When $\varphi =\varphi_{\gamma }$ then $\tilde{\varphi }=\varphi_{1-\gamma }$ as follows by substitution. Pairs $\left(\varphi_{\gamma },\varphi_{1-\gamma }\right)$ are therefore *conjugate pairs*. Inside the Cressie-Read family, the Hellinger divergence function is self-conjugate.

For $P=P_{\theta }$ and $Q\in S^{K}$ we denote $\phi \left(Q,P\right)$ by $\phi \left(Q,\theta \right)$ (resp $\phi \left(\theta ,Q\right)$, or $\phi \left(\theta^{\prime },\theta \right),$ etc. according to the context).

#### 1.1.2. Weights

This paragraph introduces the special link which connects CR divergences with specific random variables, which we call weights. Those will be associated to the dataset and define what is usually referred to as a generalized bootstrap procedure. This is the setting which allows for an interpretation of the MDE’s as generalized bootstrapped MLEs.

For a given real valued random variable (RV) *W* denote
(3)$$M\left(t\right):=\log E\left[\exp(tW)\right]$$
its cumulant generating function which we assume to be finite in a non-void open neighborhood of 0. The Fenchel Legendre transform of *M* (also called the Chernoff function) is defined through
(4)$$\varphi^{W}\left(x\right)=M^{*}\left(x\right):=\underset{t}{\sup}\left(tx-M(t)\right).$$

The function $x\to \varphi^{W}\left(x\right)$ is non-negative, is $C^{\infty }$ and convex. We also assume that EW=1 together with VarW=1 which implies $\varphi^{W}\left(1\right)={\left(\varphi^{W}\right)}^{\prime }\left(1\right)=0$ and ${\left(\varphi^{W}\right)}^{\prime \prime }\left(1\right)=1.$ Hence $\varphi^{W}\left(x\right)$ is a divergence function with corresponding divergence $\phi^{W}.$ Associated with $\varphi^{W}$ is the conjugate divergence $\tilde{\phi^{W}}$ with divergence function $\tilde{\varphi^{W}}$, which therefore satisfies $\phi^{W}\left(Q,P\right)=\tilde{\phi^{W}}\left(P,Q\right)$ whenever neither *P* nor *Q* have null entries.

It is of interest to note that the classical power divergences $\varphi_{\gamma }$ can be represented through (Equation 4) for γ≤1 or γ≥2. A first proof of this lays in the fact that when *W* has a distribution in a Natural Exponential Family (NEF) with power variance function with exponent α=2−γ, then the Legendre transform $\varphi^{W}$ of its cumulant generating function *M* is indeed of the form (Equation 1). See [16,17] for NEF’s and power variance functions, and [18] for relation to the bootstrap. A general result of a different nature, including the former ones, can be seen in [19], Theorem 20. Correspondence between the various values of γ and the distribution of the respective weights can be found in [19], Example 39, and it can be summarized as presented now.

For γ<0 the RV *W* is constructed as follows: Let *Z* be an auxiliary RV with density $f_{Z\;}$ and support [0,∞) of a stable law with parameter triplet $\left(-\frac{\gamma }{1-\gamma },0,\frac{{\left(1-\gamma \right)}^{-\gamma //(1-\gamma )}}{\gamma }\right)$ in terms of the “form B notation” on p 12 in [20]; then *W* has an absolutely continuous distribution with density
$$f_{W}\left(y\right):=\frac{\exp\left(-y/(1-\gamma \right))}{\exp(1/\gamma )}f_{Z}\left(y\right)1_{\left[0,\infty \right)}\left(y\right).$$

For γ=0 (which amounts to consider the limit as γ→0 in (Equation 1)) then *W* has a standard exponential distribution E(1) on [0,∞).

For $\gamma \in \left(0,1\right)$ then *W* has a compound Gamma-Poisson distribution
$$C\left(POI(\theta ),GAM(\alpha ,\beta )\right)$$
where
$$\theta =\frac{1}{\gamma },\alpha =\frac{1}{1-\gamma },\beta =\frac{\gamma }{1-\gamma }.$$

For γ=1, *W* has a Poisson distribution with parameter 1,POI(1).

For γ=2, the RV *W* has normal distribution with expectation and variance equal to 1.

For γ>2, the RV *W* is constructed as follows: Let *Z* be an auxiliary RV with density $f_{Z\;}$ and support (−∞,∞) of a stable law with parameter triplet $\left(\frac{\gamma }{\gamma -1},0,\frac{{\left(\gamma -1\right)}^{-\gamma //(\gamma -1)}}{\gamma }\right)$ in terms of the “form B notation” on p 12 in [20]; then *W* has an absolutely continuous distribution with density
$$f_{W}\left(y\right):=\frac{\exp\left(y/(\gamma -1\right))}{\exp(1/\gamma )}f_{Z}\left(-y\right)\;\;\;\;,y\in R.$$

## 2. Maximum Likelihood under Finitely Supported Distributions and
Simple Sampling

### 2.1. Standard Derivation

Let $X_{1},\ldots X_{n}$ be a set of *n* independent random variables with common probability measure $P_{\theta_{T}}$ and consider the Maximum Likelihood estimator of $\theta_{T}$. A common way to define the ML paradigm is as follows: For any θ consider independent random variables $\left(X_{1,\theta },\ldots X_{n,\theta }\right)$ with probability measure $P_{\theta }$, thus *sampled in the same way as the*$X_{i}$’*s*, but under some alternative θ.

Denote
$$P_{n}:=\frac{1}{n}\sum_{i=1}^{n}\delta_{X_{i}}$$
and
$$P_{n,\theta }:=\frac{1}{n}\sum_{i=1}^{n}\delta_{X_{i,\theta }}$$
the empirical measures pertaining respectively to $\left(X_{1},\ldots X_{n}\right)$ and $\left(X_{1,\theta },\ldots X_{n,\theta }\right).$

Define $\theta_{ML}$ as the value of the parameter θ for which the probability that, up to a permutation of the order of the $X_{i,\theta }$’s, the probability that $\left(X_{1,\theta },\ldots X_{n,\theta }\right)$ coincides with $X_{1},\ldots X_{n}$ is maximal, conditionally on the observed sample $X_{1},\ldots X_{n}.$ In formula
(5)$$\theta_{ML}:=\arg\max_{\theta }P_{\theta }\left(\left.P_{n,\theta }=P_{n}\right|P_{n}\right).$$

An explicit enumeration of the above expression $P_{\theta }\left(\left.P_{n,\theta }=P_{n}\right|P_{n}\right)$ involves the quantities
$$n_{j}:=card\left\{i:X_{i}=d_{j}\right\}$$
for j=1,…,K and yields
(6)$$P_{\theta }\left(\left.P_{n,\theta }=P_{n,X}\right|P_{n,X}\right)=\frac{n!P_{\theta }{\left(d_{j}\right)}^{n_{j}}}{\prod_{j=1}^{K}n_{j}!}$$
as follows from the classical multinomial distribution. Optimizing on θ in (Equation 6) yields
$$\begin{array}{cc} \theta_{ML} & =\arg\max_{\theta }\sum_{j=1}^{K}\frac{n_{j}}{n}\log P_{\theta }\left(d_{j}\right)\\ \; & =\arg\max_{\theta }\frac{1}{n}\sum_{i=1}^{n}\log P_{\theta }\left(X_{i}\right). \end{array}$$

It follows from direct evaluation that
$$\theta_{ML}=\arg\underset{\theta }{\inf}KL_{m}\left(P_{\theta },P_{n}\right).$$

Introducing the Kullback-Leibler divergence $KL(P_{n},P_{\theta })$ it thus holds
$$\theta_{ML}=\arg\underset{\theta }{\inf}\tilde{KL_{m}}\left(P_{n},P_{\theta }\right)=\arg\underset{\theta }{\inf}KL\left(P_{n},P_{\theta }\right).$$

We have recalled that minimizing the Kullback-Leibler divergence $KL\left(P_{n},\theta \right)$ amounts to minimizing the Likelihood divergence $KL_{m}\left(\theta ,P_{n}\right)$ and produces the ML estimate of $\theta_{T}.$

### 2.2. Asymptotic Derivation

We assume that
$$\lim_{n\to \infty }P_{n}=P_{\theta_{T}}\;\;\;\;a.s.$$

This holds for example when the $X_{i}$’s are drawn as an i.i.d. sample with common law $P_{\theta_{T}}$ which we may assume in the present context. From an asymptotic standpoint, Kullback-Leibler divergence is related to the way $P_{n}$ keeps away from $P_{\theta }$ when θ is not equal to the true value of the parameter $\theta_{T}$ generating the observations $X_{i}$’s and is closely related with the type of sampling of the $X_{i}$’s. In the present case, when i.i.d. sampling of the $X_{i,\theta }$’s under $P_{\theta }$ is performed, Sanov Large Deviation theorem leads to
(7)$$\lim_{n\to \infty }\frac{1}{n}\log P_{\theta }\left(\left.P_{n,\theta }=P_{n}\right|P_{n}\right)=-KL\left(\theta_{T},\theta \right).$$

This result can easily be obtained from (Equation 6) using Stirling formula to handle the factorial terms and the law of large numbers which states that for all *j*’s, $n_{j}/n$ tends to $P_{\theta_{T}}\left(d_{j}\right)$ as *n* tends to infinity. We note that the MLE $\theta_{ML}$ is a proxy of the minimizer of the natural estimator $\theta_{T}$ of $KL\left(\theta_{T},\theta \right)$ in θ, substituting the unknown measure generating the $X_{i}$’s by its empirical counterpart $P_{n}$. Alternatively as will be used in the sequel, $\theta_{ML}$ minimizes upon θ the Likelihood divergence $KL_{m}\left(\theta ,\theta_{T}\right)$ between $P_{\theta }$ and $P_{\theta_{T}}$ substituting the unknown measure $P_{\theta_{T}}$ generating the $X_{i}$’s by its empirical counterpart $P_{n}$. Summarizing we have obtained:

The ML estimate can be obtained from a LDP statement as given in (Equation 7), optimizing in θ in the estimator of the LDP rate where the plug-in method of the empirical measure of the data is used instead of the unknown measure $P_{\theta_{T}}.$ Alternatively it holds
(8)$$\theta_{ML}:=\arg\min_{\theta }\hat{KL_{m}}\left(\theta ,\theta_{T}\right)$$
with
$$\hat{KL_{m}}\left(\theta ,\theta_{T}\right):=KL_{m}\left(\theta ,P_{n}\right).$$

This principle will be kept throughout this paper: the estimator is defined as maximizing the probability that the simulated empirical measure be close to the empirical measure as observed on the sample, conditionally on it, following the same sampling scheme. This yields a maximum likelihood estimator, and its properties are then obtained when randomness is introduced as resulting from the sampling scheme.

## 3. Bootstrap and Weighted Sampling

The sampling scheme which we consider is commonly used in connection with the bootstrap and is referred to as the *weighted* or *generalized bootstrap*, sometimes called *wild bootstrap*, first introduced by Newton and Mason [21].

Let $X_{1},\ldots ,X_{n}$ with common distribution *P* on $Y:=\left\{d_{1},\ldots ,d_{K}\right\}.$

Consider a collection $W_{1},\ldots ,W_{n}$ of independent copies of *W*, whose distribution satisfies the conditions stated in Section 1. The weighted empirical measure $P_{n}^{W}$ is defined through
$$P_{n}^{W}:=\frac{1}{n}\sum_{i=1}^{n}W_{i}\delta_{X_{i}}.$$

This empirical measure need not be a probability measure, since its mass may not equal 1. Also it might not be positive, since the weights may take negative values. Therefore $P_{n}^{W}$ can be identified with a random point in $R^{K}.$ The measure $P_{n}^{W}$ converges almost surely to *P* when the weights $W_{i}$’s satisfy the hypotheses stated in Section 1.

We also consider the normalized weighted empirical measure
(9)$$P_{n}^{W}:=\sum_{i=1}^{n}Z_{i}\delta_{X_{i}}$$
where
(10)$$Z_{i}:=\frac{W_{i}}{\sum_{j=1}^{n}W_{j}}$$
whenever $\sum_{j=1}^{n}W_{j}\neq 0$, and
$$P_{n}^{W}=\infty$$
when $\sum_{j=1}^{n}W_{j}=0$, where $P_{n}^{W}=\infty$ means $P_{n}^{W}\left(d_{k}\right)=\infty$ for all $d_{k}$ in Y.

### 3.1. A Conditional Sanov Type Result for the Weighted Empirical Measure

We now state a conditional Sanov type result for the family of random measures $P_{n}^{W}.$ It follows readily from a companion result pertaining to $P_{n}^{W}$ and enjoys a simple form when the weights $W_{i}$ are associated to power divergences, as defined in Section 1.1.2. We quote the following results, referring to [19].

Consider a set Ω in $R^{K}$ such that
(11)$$cl\Omega =cl[\mathrm{Int}\left(\Omega \right)]$$
which amounts to a regularity assumption (obviously met when Ω is an open set), and which allows for the replacement of the usual liminf and limsup by standard limits in usual LDP statements. We denote by $P^{W}$ the probability measure of the random family of i.i.d. weights $W_{i}.$

It then holds

**Proposition** **1**(Theorem 9 in [19])**.**
*The weighted empirical measure $P_{n}^{W}$ satisfies a conditional Large Deviation Principle in $R^{K}$ namely, denoting P the a.s. limit of $P_{n},$*
$$\lim_{n\to \infty }\frac{1}{n}\log P^{W}\left(\left.P_{n}^{W}\in \Omega \right|X_{1}^{n}\right)=-\phi^{W}\left(\Omega ,P\right)$$
*where $\phi^{W}\left(\Omega ,P\right):=\inf_{Q\in \Omega }\phi^{W}\left(Q,P\right).$*


As a direct consequence of the former result, it holds, for any $\Omega \subset S^{K}$ satisfying (Equation 11), where $S^{K}$ designates the simplex of all pm’s on Y.

**Theorem** **1**(Theorem 12 in [19])**.**
*The normalized weighted empirical measure $P_{n}^{W}$ satisfies a conditional Large Deviation Principle in $S^{K}$*
(12)$$\lim_{n\to \infty }\frac{1}{n}\log P^{W}\left(\left.P_{n}^{W}\in \Omega \right|X_{1}^{n}\right)=-\underset{m\neq 0}{\inf}\phi^{W}\left(m\Omega ,P\right).$$


A flavour of the simple proofs of Proposition 1 and Theorem 1 is presented in Appendix A; see [19] for a detailed treatment; see also Theorem 3.2 and Corollary 3.3 in [22] where Theorem 1 is proved in a more abstract setting.

We will be interested in the pm’s in Ω which minimize the RHS in the above display. The case when $\;\phi^{W}$ is a power divergence, namely $\phi^{W}=\phi_{\gamma }$ for some γ enjoys a special property with respect to the pm’s *Q* achieving the infimum (upon *Q* in Ω) in (Equation 12). It holds

**Proposition** **2**(Lemma 14 in [19])**.**
*Assume that $\phi^{W}$ is a power divergence. Then*
$$Q\in \arg\inf\left\{\underset{m\neq 0}{\inf}\phi^{W}\left(mQ,P\right),Q\in \Omega \right\}$$
*and*
$$Q\in \arg\inf\left\{\phi^{W}\left(Q,P\right),Q\in \Omega \right\}$$
*are equivalent statements.*


Indeed Proposition 2 holds as a consequence of the following results, to be used later on.

**Lemma** **1.**
*For Q and P two pm’s such that the involved expressions are finite, it holds*
*(i)* 
*For γ≠0 and γ≠1 it holds that*
$$\underset{m\neq 0}{\inf}\phi_{\gamma }\left(mQ,P\right)=\frac{1}{\gamma }\left[1-{\left(1+\gamma \left(\gamma -1\right)\phi_{\gamma }\left(Q,P\right)\right)}^{-1/(\gamma -1)}\right].$$
*(ii)* )
$\inf_{m\neq 0}\phi_{1}\left(mQ,P\right)=1-\exp\left(-KL(Q,P)\right)=1-\exp\left(-\phi_{1}\left(Q,P\right)\right).$
*(iii)* 
$\inf_{m\neq 0}\phi_{0}\left(mQ,P\right)=KL_{m}\left(Q,P\right)=\phi_{0}\left(Q,P\right)$



In the case where *W* is a RV with standard exponential distribution, then a link between the present approach and Bayesian inference can be drawn, since the normalized weighted empirical measure $P_{n}^{W}$ is a realization of the a posteriori distribution for the Dirichlet prior on the non parametric distribution of X. See [23].

The weighted empirical measure $P_{n}^{W}$ has been used in the weighted bootstrap (or wild bootstrap) context, although it is not a pm. However, conditionally upon the sample points, its produces statistical estimators $T(P_{n}^{W})$ whose weak behavior (conditionally upon the sample) converges to the same limit as does $T(P_{n})$ when normalized on the classical CLT range; see eg Newton and Mason [21]. Large deviation theorem for the weighted empirical measure $P_{n}^{W}$ has been obtained by [24]; for other contributions in line with those, see [22,25]. Normalizing the weights produces families of exchangeable weights $Z_{i}$, and the normalized weighted empirical measure $P_{n}^{W}$ is the cornerstone for the so-called non-parametric Bayesian bootstrap, initiated by [23], and further developed by [26] among others. Note however that in this context the RV’s $W_{i}$’s are chosen distributed as standard exponential variables. The link with spacings from a uniform distribution and the corresponding reproducibility of the Dirichlet distributions are the basic ingredients which justify the non parametric bootstrap approach; in the present context, the choice of the distribution of the $W_{i}$’s is a natural extension of this paradigm, at least when those $W_{i}$’s are positive RV’s.

### 3.2. Maximum Likelihood for the Generalized Bootstrap

Let’s turn back to the estimation of $\theta_{T}$, assuming $P_{\theta_{T}}$ the common distribution of the independent observations $X_{1},\ldots ,X_{n}$. We will consider maximum likelihood in the same spirit as developed in Section 2.2, here in the context of the normalized weighted empirical measure; it amounts to justify minimum divergence estimators as appropriate MLEs under such bootstrap procedure.

We thus consider the same statistical model $P_{\Theta }$ and keep in mind the ML principle as seen as resulting from a maximization of the conditional probability of getting simulated observations close to the initially observed data. Similarly as in Section 2 fix an arbitrary θ and simulate $X_{1,\theta },\ldots ,X_{n,\theta }$ with distribution $P_{\theta }.$ Define accordingly $P_{n,\theta }^{W}$ and $P_{n,\theta }^{W}$ making use of i.i.d. RV’s $W_{1},\ldots ,W_{n}$. Now the event $P_{n,\theta }^{W}\left(k\right)=n_{k}/n$ has probability 0 in most cases (for example when *W* has a continuous distribution), and therefore we are led to consider events of the form $P_{n,\theta }^{W}\in V_{\varepsilon }\left(P_{n}\right)$, meaning $\max_{k}\left|P_{n,\theta }^{W}\left(d_{k}\right)-P_{n}\left(d_{k}\right)\right|\leq \varepsilon$ for some ε>0; notice that $V_{\varepsilon }\left(P_{n}\right)$ defined through
$$V_{\varepsilon }\left(P_{n}\right):=\left\{Q\in S^{K}:\max_{k}\left|Q\left(d_{k}\right)-P_{n}\left(d_{k}\right)\right|\leq \varepsilon \right\}$$
has non-void interior.

For such a configuration consider
(13)$$P^{W}\left(\left.P_{n,\theta }^{w}\in V_{\varepsilon }\left(P_{n}\right)\right|X_{1,\theta },\ldots ,X_{n,\theta },P_{n}\right)$$
where the $X_{i,\theta }$ are randomly drawn i.i.d. under $P_{\theta }.$ Obviously for θ far away from $\theta_{T}$ the sample $\left(X_{1,\theta },\ldots ,X_{n,\theta }\right)$ is realized “far away ” from $\left(X_{1},\ldots ,X_{n}\right)$, which has been generated under the truth, namely $P_{\theta_{T}}$, and the probability in (Equation 13) is small, whatever the weights, for small ε.

We will now consider (Equation 13) for large *n*, since, in contrast with the first derivation of the standard MLE in Section 2.1, we cannot perform the same calculation for each *n*, which was based on multinomial counts. Note that we obtained a justification for the usual MLE through the asymptotic Sanov LDP, leading to the KL divergence and finally back to the MLE through an approximation step of this latest. From Theorem Equation 12 together with the a.s. convergence of $P_{n}$ to $P_{\theta_{T}}$ in $S^{K}$ it follows that for some α<1<β
(14)$$\begin{array}{cc} \; & -\underset{m\neq 0}{\inf}\phi^{W}\left(mV_{\alpha \epsilon }\left(P_{\theta_{T}}\right),\theta \right)\\ \; & \leq \lim_{n\to \infty }\frac{1}{n}\log P^{W}\left(P_{n,\theta }^{W}\in V_{\epsilon }\left(P_{n}\right)|X_{1,\theta },\ldots ,X_{n,\theta },P_{n}\right)\\ \; & \leq -\underset{m\neq 0}{\inf}\phi^{W}\left(mV_{\beta \epsilon }\left(P_{\theta_{T}}\right),\theta \right) \end{array}$$
where $\phi^{W}\left(V_{c\epsilon }\left(\theta_{T}\right),\theta \right)=\inf_{\mu \in V_{c\epsilon }\left(P_{\theta_{T}}\right))}\phi^{W}\left(\mu ,\theta \right)$.

As ε→0, by continuity it holds that
(15)$$\begin{array}{ccc} & & \lim_{\varepsilon \to 0}\lim_{n\to \infty }\frac{1}{n}\log P^{W}\left(P_{n,\theta }^{W}\in V_{\epsilon }\left(P_{n}\right)|X_{1,\theta },\ldots ,X_{n,\theta },P_{n}\right)\\ & = & -\underset{m\neq 0}{\inf}\phi^{W}\left(mP_{\theta_{T}},\theta \right). \end{array}$$

The ML principle amounts to maximize
(16)$$P^{W}\left(P_{n,\theta }^{W}\in V_{\epsilon }\left(P_{n}\right)|X_{1,\theta },\ldots ,X_{n,\theta },P_{n}\right)$$
over θ. Whenever Θ is a compact set we may insert this optimization in (Equation 14) which yields, following (Equation 15)
$$\begin{array}{ccc} & & \lim_{\varepsilon \to 0}\lim_{n\to \infty }\frac{1}{n}\log\underset{\theta }{\sup}P^{W}\left(P_{n,\theta }^{W}\in V_{\epsilon }\left(P_{n}\right)|X_{1,\theta },\ldots ,X_{n,\theta },P_{n}\right)\\ & = & -\underset{\theta \in \Theta }{\inf}\underset{m\neq 0}{\inf}\phi^{W}\left(mP_{\theta_{T}},\theta \right). \end{array}$$

We consider weights *W*’s such that there exists a power divergence function $\varphi_{\gamma }$ satisfying (Equation 4), which amounts to $\phi^{W}=\phi_{\gamma }$; by the results quoted in Section 1.1.2 this holds when γ∈(−∞,1]∪[2,+∞).

By Proposition 2 the argument of the infimum upon θ in the RHS of the above display coincides with the corresponding argument of $\phi^{W}\left(\theta_{T},\theta \right)$, which obviously gets $\theta_{T}$. This justifies to consider a proxy of this minimization problem as a “ML” estimator based on normalized weighted data.

A further interpretation of the MDE in the context of non-parametric Bayesian procedures may also be proposed; this is postponed to a next paper.

Since
$$\phi^{W}\left(\theta_{T},\theta \right)={\tilde{\phi }}^{W}\left(\theta ,\theta_{T}\right)$$
the ML estimator is obtained as in the conventional case by plug in the LDP rate. Obviously the “best” plug in consists in the substitution of $P_{\theta_{T}}$ by $P_{n},$ the empirical measure of the sample, since $P_{n}$ achieves the best rate of convergence to $P_{\theta_{T}}$ when confronted to any bootstrapped version, which adds “noise” to the sampling. We may therefore call
(17)$$\begin{array}{cc} \theta_{ML}^{W} & :=\arg\underset{\theta \in \Theta }{\inf}{\tilde{\phi }}^{W}\left(\theta ,P_{n}\right):=\arg\underset{\theta \in \Theta }{\inf}\sum_{k=1}^{K}P_{n}\left(d_{k}\right){\tilde{\varphi }}^{W}\left(\frac{P_{\theta }\left(d_{k}\right)}{P_{n}\left(d_{k}\right)}\right)\\ \; & =\arg\underset{\theta \in \Theta }{\inf}\sum_{k=1}^{K}P_{\theta }\left(d_{k}\right)\varphi^{W}\left(\frac{P_{n}\left(d_{k}\right)}{P_{\theta }\left(d_{k}\right)}\right) \end{array}$$
the MLE for the bootstrap sampling; here ${\tilde{\phi }}^{W}$ (with divergence function $\tilde{\varphi }$) is the conjugate divergence of $\phi^{W}$ (with divergence function φ). Since $\phi^{W}=\phi_{\gamma }$ for some γ, it holds ${\tilde{\phi }}^{W}=\phi_{1-\gamma }.$

We can also plug in the normalized weighted empirical measure, which also is a proxy of $P_{\theta_{T}}$ for each run of the weights. This produces a bootstrap estimate of $\theta_{T}$ through
(18)$$\begin{array}{cc} \theta_{B}^{W} & :=\arg\underset{\theta \in \Theta }{\inf}{\tilde{\phi }}^{W}\left(\theta ,P_{n}^{W}\right):=\arg\underset{\theta \in \Theta }{\inf}\sum_{k=1}^{K}P_{n}^{W}\left(d_{k}\right){\tilde{\varphi }}^{W}\left(\frac{P_{\theta }\left(d_{k}\right)}{P_{n}^{W}\left(d_{k}\right)}\right)\\ \; & =\arg\underset{\theta \in \Theta }{\inf}\sum_{k=1}^{K}P_{\theta }\left(d_{k}\right)\varphi^{W}\left(\frac{P_{n}^{W}\left(d_{k}\right)}{P_{\theta }\left(d_{k}\right)}\right) \end{array}$$
where $P_{n}^{W}$ is defined in (Equation 9), assuming *n* large enough such that the sum of the $W_{i}$’s is not zero. Whenever P(W=0)>0, these estimators are defined for large *n* in order that $P_{n}^{W}\left(d_{k}\right)$ be positive for all k. Since E(W)=1, this occurs for large samples.

For a given weighted bootstrapped sample with weights $W_{1},\ldots ,W_{n}$ leading to the weighted normalized empirical measure $P_{n}^{W}$, $\theta_{B}^{W}$ is the MLE in the sense of (Equation 16), hence defined as a proxy of the maximizer of
$$P^{W^{\prime }}\left(P_{n,\theta }^{W^{\prime }}\in V_{\epsilon }\left(P_{n}^{W}\right)|X_{1,\theta },\ldots ,X_{n,\theta },P_{n}^{W}\right)$$
where the vector $\left(W_{1}^{\prime },\ldots ,W_{n}^{\prime }\right)$ is an independent copy of $\left(W,\ldots ,W_{n}\right).$ This estimator usually differs from the bootstrapped version of the MLE based on $P_{n}$ (see (Equation 8)) which is defined for *n* large enough through
$$\theta_{ML}^{B}:=\arg\underset{\theta }{\inf}KL_{m}\left(\theta ,P_{n}^{W}\right).$$

When Y is not a finite space then an equivalent construction can be developed based on the variational form of the divergence; see [6].

**Remark** **1.**
*We may also consider cases when the MLE defined through $\theta_{ML}^{W}$ defined in (Equation 17) coincide with the standard MLE $\theta_{ML}$ under i.i.d. sampling, and when its bootstrapped counterparts $\theta_{B}^{W}$ defined in (Equation 18) coincides with the bootstrapped standard MLE $\theta_{ML}^{b}$ defined through the likelihood estimating equation where the factor 1/n is substituted by the weight $Z_{i}$. It is proved in Theorem 5 of [11] that whenever $P_{\Theta }$ is an exponential family with natural parametrization $\theta \in R^{d}$ and sufficient statistics T*
$$P_{\theta }\left(d_{j}\right)=\exp\left[T{\left(d_{j}\right)}^{\prime }\theta -C\left(\theta \right)\right],\;\;\;\;1\leq j\leq K$$
*where the Hessian matrix of C(θ) is definite positive, then for all divergence pseudo distance ϕ satisfying regularity conditions (including therefore the present cases), $\theta_{ML}^{W}$ equals $\theta_{ML}$, the classical MLE in $P_{\Theta }$ defined as the solution of the normal equation*
$$\frac{1}{n}\sum T\left(X_{i}\right)=\nabla C\left(\theta_{ML}\right)$$
*irrespectively upon ϕ. Therefore on regular exponential families, and under i.i.d. sampling, all minimum divergence estimators coincide with the MLE (which is indeed one of them). The proof of this result is based on the variational form of the estimated divergence $Q\to \phi \left(Q,P\right)$, which coincides with the plug in version in (Equation 17) when the common support of all distributions in $P_{\Theta }$ is finite. Following verbatim the proof of Theorem 5 in [11] substituting $P_{n}$ by $P_{n}^{W}$ it results that $\theta_{B}^{W}$ equals the weighted MLE (standard generalized bootstrapped MLE $\theta_{ML}^{b}$) defined through the normal equation*
$$\sum_{i=1}^{n}Z_{i}T\left(X_{i}\right)=\nabla C\left(\theta_{ML}^{b}\right),$$
*where the $Z_{i}$’s are defined in (Equation 10). This fact holds for any choice of the weights, irrespectively on the choice of the divergence function φ with the only restriction that it satisfies the mild conditions (RC) in [11]. It results that for those models any generalized bootstrapped MDE coincides with the corresponding standard bootstrapped MLE.*


**Example** **1.**
*A-When W has a standard Poisson POI(1) distribution then the resulting estimator is the minimum modified Kullback-Leibler one. which takes the usual weighted form of the standard generalized bootstrap MLE*
$$\theta_{B}^{POI(1)}:=\arg\underset{\theta }{\sup}\sum_{k=1}^{K}\left(\frac{\sum_{i=1}^{n}W_{i}1_{k}\left(X_{i}\right)}{\sum_{i=1}^{n}W_{i}}\right)\log P_{\theta }\left(k\right)$$
*which is defined for n large enough. Also in this case $\theta_{ML}^{W}$ coincides with the standard MLE.*

*B-If W has an Inverse Gaussian distribution IG(1,1) then $\;\varphi \left(x\right)=\varphi_{-1}\left(x\right)=\frac{1}{2}{\left(x-1\right)}^{2}/x$ for x>0 and the ML estimator minimizes the Pearson Chi-square divergence with generator function $\varphi_{2}\left(x\right)=\frac{1}{2}{\left(x-1\right)}^{2}$ which is defined on R.*

*C-If W follows a normal distribution with expectation and variance 1, then the resulting divergence is the Pearson Chi-square divergence $\varphi_{2}\left(x\right)$ and the resulting estimator minimizes the Neyman Chi-square divergence with $\varphi \left(x\right)=\varphi_{-1}\left(x\right).$*

*D-When W has a Compound Poisson Gamma distribution $C\left(POI(2),\Gamma (2,1)\right)$ distribution then the corresponding divergence is $\varphi_{1/2}\left(x\right)=2{\left(\sqrt{x}-1\right)}^{2}$ which is self conjugate, whence the ML estimator is the minimum Hellinger distance one.*


## 4. Bahadur Efficiency of Minimum Divergence Tests under Generalized Bootstrap

In [27] Efron and Tibshirani suggest the bootstrap as a valuable approach for testing, based on bootstrapped samples. We show that bootstrap testing for parametric models based on appropriate divergence statistics enjoys maximal Bahadur efficiency with respect to any bootstrap test statistics.

The standard approach to Bahadur efficiency can be adapted for the present generalized Bootstrapped tests as follows.

Consider the test of some null hypothesis H0: $\theta_{T}=\theta$ versus a simple Hypothesis H1 $\theta_{T}=\theta^{\prime }.$

We consider two competitive statistics for this problem. The first one is based on the bootstrap estimate of ${\tilde{\phi }}^{W}\left(\theta ,\theta_{T}\right)$ and
$$T_{n,X}:=\tilde{\Phi }\left(\theta ,P_{n,X}^{W}\right)=T\left(P_{n,X}^{W}\right)$$
which allows to reject H0 for large values since $\lim_{n\to \infty }$$T_{n,X}=0$ whenever H0 holds. In the above display we have emphasized in $P_{n,X}^{W}$ the fact that we have used the RV $X_{i}$’s. Let
$$L_{n}\left(t\right):=P^{W}\left(\left.T_{n,X}>t\right|X_{1},\ldots ,X_{n}\right).$$

We use $P^{W}$ to emphasize that the hazard is due to the weights. Consider now a set of RVs $Z_{1},\ldots ,Z_{n}$ extracted from a sequence such that
$$\lim_{n\to \infty }P_{n,Z}=P_{\theta^{\prime }}$$
a.s; we have denoted $P_{n,Z}$ the empirical measure of $\left(Z_{1},\ldots ,Z_{n}\right);$ accordingly define $P_{n,Z}^{W^{\prime }},$ the normalized weighted empirical measure of the $Z_{i}$ ’s making use of weights $\left(W_{1}^{\prime },\ldots ,W_{n}^{\prime }\right)$ which are i.i.d. copies of $\left(W_{1},\ldots ,W_{n}\right)$, drawn independently from $\left(W_{1},\ldots ,W_{n}\right).$ Define accordingly
$$T_{n,Z}:=\tilde{\Phi }\left(\theta ,P_{n,Z}^{W^{\prime }}\right)=T\left(P_{n,Z}^{W^{\prime }}\right).$$

Define
$$L_{n}\left(T_{n,Z}\right):=P^{W}\left(\left.T_{n,W}>T_{n,Z}\right|X_{1},\ldots ,X_{n}\right)$$
which is a RV (as a function of $T_{n,Z}$). It holds
$$\lim_{n\to \infty }T_{n,Z}=\tilde{\Phi }\left(\theta ,\theta^{\prime }\right)\;\;\;\;a.s$$
and therefore the Bahadur slope for the test with statistics $T_{n}$ is $\Phi \left(\theta^{\prime },\theta \right)$ as follows from
$$\begin{array}{cc} \lim_{n\to \infty }\frac{1}{n}\log L_{n}\left(T_{n,Z}\right) & =-\inf\left\{\Phi \left(Q,\theta_{T}\right):\tilde{\Phi }\left(\theta ,Q\right)>\tilde{\Phi }\left(\theta ,\theta^{\prime }\right)\right\}\\ \; & =-\inf\left\{\Phi \left(Q,\theta_{T}\right):\Phi \left(Q,\theta \right)>\Phi \left(\theta^{\prime },\theta \right)\right\}\\ \; & =-\Phi \left(\theta^{\prime },\theta \right) \end{array}$$

If $\theta_{T}=\theta .$ Under H0 the rate of decay of the *p*-value corresponding to a sampling under H1 is captured through the divergence $\Phi \left(\theta^{\prime },\theta \right).$

Consider now a competitive test statistics $S\left(P_{n,X}^{W}\right)$ and evaluate its Bahadur slope. Similarly as above it holds, assuming continuity of the functional *S* on $S^{K}$
$$\begin{array}{cc} \; & \lim_{n\to \infty }\frac{1}{n}\log P^{W}\left(\left.S\left(P_{n,X}^{W}\right)>S\left(P_{n,Z}^{W^{{}^{\prime }}}\right)\right|X_{1},\ldots ,X_{n}\right)\\ \; & =-\inf\left\{\Phi \left(Q,\theta_{T}\right):S\left(Q\right)>S\left(\theta^{\prime }\right)\right\}\\ \; & \geq -\Phi \left(\theta^{\prime },\theta_{T}\right) \end{array}$$
as follows from the continuity of $Q\to \Phi \left(Q,\theta_{T}\right).$ Hence the Bahadur slope of the test based on $S\left(P_{n,X}^{W}\right)$ is larger or equal $\Phi \left(\theta^{\prime },\theta \right).$

We have proved that the chances under H0 for the statistics $T_{n,X}$ to exceed a value obtained under H1 are (asymptotically) less that the corresponding chances associated with any other statistics based on the same bootstrapped sample; as such it is most specific on this scale with respect to any competing ones. Namely the following result holds:

**Proposition** **3.**
*Under the weighted sampling the test statistics $T\left(P_{n,X}^{W}\right)$ is the most efficient among all tests which are empirical versions of continuous functionals on $S^{K}.$*

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
