# Peer review of "Minimum Divergence Estimators, Maximum Likelihood and the Generalized Bootstrap"

_entropy, 2021, doi:10.3390/e23020185_

Round 1
Reviewer 1 Report
Please find attached.

Reviewer 2 Report
Please see attached file.

Round 2
Reviewer 1 Report
I feel the introduction can be little more clearer. For example, in page 2, 4th paragraph, I don't understand how the second sentence "This limit statement is nothing but..." is related to the first sentence in that paragraph.
In the last but one paragraph in page 2, it should be, "Due to the type of asymptotics handled..."
In page 5, in the definition of \theta_{ML}, in the summation over, that symbol must be explained.
I am not sure if the derivation in Sec. 2.1 that the MLE is the minimizer of the KL divergence in the finite support case necessary as that is very well-known.
Author Response
The paper has been changed according to the requirements of the referees.
With respect to the comments of the first referee, the section on the well known standard likelihood estimator has been shortened ,just leaving the essential step of the derivation, which serves as a benchmark for the asymptotic derivation, which in turn is the basis for the present construction. Misprints and typos have been corrected, and some comments have been withdrawn, for clearity and in order to keep the paper as self consistent as possible. To state it plainly, I think that it would be difficult to present a similar construction outside the range of power divergences, or divergences which can be reduced to them (Rény type, Jensen Shannon, etc).
Reviewer 2 Report
I have no other technical concerns. However, the paper is written in a loose, and somewhat sloppy manner, which is not in conformity with the technical quality of the paper.
I have chosen "minor revision", instead of "Accept in present form", because I really believe that while the paper is technically good, the paper needs a fair bit of improved organization in style.
There are too many typos, misplaced, missing or incorrect punctuation marks, unnecessary paragraph breaks, unnecessary capital letters, and even incomplete sentences. In short, the paper needs a very careful proofreading and polishing before it conforms to the requirements of the journal.
Some examples:
1. Three lines before the start of Section 5 (Appendix). A sentence starts with "Namely ...", but then just drops offs and continues to the next para without finishing the sentence.
2. The sections of the Appendix are numbered as 5.0.1, ... etc. Why not simply 5.1 ... etc.
3. After the paper ends in page 20, the next line has the word "References" in regular font. Two lines later "References" again in bold and larger font. The first one must be superfluous.
I like the paper otherwise, so I am unhappy to see that presentation is not up to the mark. I had noted the necessity of this in the previous version also. I hope the author takes care of these before publication.
Author Response
The paper has been changed according to the requirements of the referees.
The second referee pointed a number of misprints and asked for a careful check on the presentation of the paper ; I have corrected them and I think that the present version meets his requirements.